# Characteristics and Source of Dissolved Organic Matter in Lake Hulun, A Large Shallow Eutrophic Steppe Lake in Northern China

**Wenwen Wang** [1,2,3], **Binghui Zheng** [1,2,3], **Xia Jiang** [1,2,3], **Junyi Chen** [1,2] and **Shuhang Wang** [1,2,*]

[1]  National Engineering Laboratory for Lake Pollution Control and Ecological Restoration, Chinese Research Academy of Environmental Sciences, Beijing 100012, China; wangwenwen-5720088@163.com (W.W.); zhengbinghui@craes.org.cn (B.Z.); jiangxia@craes.org.cn (X.J.); beyond8090@163.com (J.C.)

[2]  State Environment Protection Key Laboratory for Lake Pollution Control, Chinese Research Academy of Environmental Sciences, Beijing 100012, China

[3]  College of Water Sciences, Beijing Normal University, Beijing 100875, China

*  Correspondence: shuhang125126@163.com; Tel.: +86-010-84913896

**Abstract:** Lake Hulun, the fifth largest lake in China, is a typical eutrophic steppe lake located in the Hulun Buir Prairie. The dissolved organic matter (DOM) in the water of Lake Hulun has a high concentration. However, little is known about the occurrence characteristics and source of the DOM in Lake Hulun. The spatial and temporal distribution characteristics of DOM concentration in Lake Hulun were thoroughly surveyed, and the optical characteristics, fluorescence components and sources of DOM were analyzed by excitation emission matrix (EMM) and parallel factor analysis (PARAFAC) technology. The DOM concentration was 6.46–42.87 mg C/L, and was highest in summer and lowest in winter. The difference in the spatial distribution of DOM in winter was significant due to the ice over, and showed a trend where the concentration near the shore was higher than that in the center of the lake. Three humic-like components and one component consisting of a mixture of humic-like and protein-like substances of DOM were identified, with the former being prevalent. The humification index of DOM was 2.22–9.92, indicating that the DOM has a high degree of humification. The DOM is mainly derived from terrestrial sources, with the highest proportion (91.0% ± 8.1%) found in winter and the lowest (66.2% ± 7.7%) in summer. Given that the DOM in Lake Hulun is mainly dominated by humic-like components with a high degree humification, the DOM may have low bioactivity. However, this is just a preliminary analysis and judgment, and it is necessary to conduct other experiments such as biodegradation experiments to further study the bioavailability of DOM in Lake Hulun.

**Keywords:** steppe lake; global warming; Lake Hulun; DOM; source

## 1. Introduction

Dissolved organic matter (DOM) is widely found in rivers, lakes, and other natural water bodies and it is considered to be the largest pool of organic matter that occurs in natural waters. As an important and active chemical component in terrestrial and aquatic ecosystems, DOM plays an important role in aquatic ecosystems and is of great significance in the biogeochemical cycle of biogenic elements, such as carbon, nitrogen, and phosphorus, in water bodies [1–3]. In recent years, studies have found a general increase in the concentration of dissolved organic carbon (DOC) in surface waters in eastern North America and northern and central Europe. Studies have also found that DOC concentrations will continue to rise, with unpredictable effects on the global carbon cycle [4]. Climate

change is considered one of the most important factors that influence DOM concentration in surface waters [5–7].

Arid and semi-arid areas account for nearly one-third of the land area of the world and half of China's land area [8]. Lakes in these areas are important inland ecosystems that play an important role in local economic development, as well as providing drinking water and being involved in transportation and entertainment [9,10]. However, lakes are highly sensitive to the effects of climate change and human activity [8,11]. In the context of global warming, the concentration, components, and sources of DOM in lake waters in arid and semi-arid regions are bound to be affected to a great extent. Lake Hulun is a typical large, shallow lake in the arid and semi-arid areas of northern China [12,13]. In recent years, the water quality of Lake Hulun has been deteriorating and the water ecology has been degraded significantly under the double influences of anthropogenic disturbance and climate change [14,15]. A previous study has shown that DOM concentration in Lake Hulun is as high as 59 mg C/L [14]. However, the spatial and temporal distribution characteristics, bioavailability, influencing factors, and future development trend of DOM in the lake under global warming are still unclear.

Excitation emission matrix (EEM) fluorescence spectroscopy is a powerful and economical tool to distinguish different groups of organic compounds in DOM samples [16]. Parallel factor analysis (PARAFAC) can decompose the complex EMM data into individual components and can also determine the relative contribution of each component to the total EEM fluorescence [17]. The EEM and PARAFAC have frequently been combined and used to characterize DOM fluorescence. The DOM concentration in Lake Hulun in spring, summer, autumn, and winter was investigated, and the EMM-PARAFAC technology was also used to study (1) the spatial and temporal distribution characteristics of DOM in Lake Hulun, (2) the optical characteristics and components of DOM, and (3) the sources of DOM and their relative contribution.

## 2. Materials and Methods

### 2.1. Study Area

Lake Hulun (48.55–49.33° N, 116.97–117.81° E), also known as Lake Dalai, is the fifth largest lake in China and the largest lake in northern China [12,13]. The Lake Hulun basin (including the Haraha River and Hailar River basins) is located in China and Mongolia with an area of $2.92 \times 10^5$ km², of which the basin area in China is $1.08 \times 10^5$ km², accounting for 37% of the total area. The main natural water inflows of Lake Hulun are the Crulen, Orshen, and Hailar rivers, and the drainage channel is the Xinkai River. The Hulun Lake basin has a temperate semi-arid continental climate with the characteristics of a temperate climate dominated by cold and heat upheavals. Spring is dry and windy, summer is cool and short, the temperature during autumn drops sharply, frost occurs early, and winter is cold and long. Given the strong atmospheric transparency and high solar radiation intensity, the annual average sunshine hours are 3104.7 h. The annual precipitation of the basin is 247–319 mm, most (80%–86%) of which occurs during June and September. The mean annual evaporation reaches 1400–1900 mm, which is 5–6 times the annual precipitation [18].

Lake Hulun plays a unique role in regulating regional climate, conserving water, preventing desertification, protecting biodiversity, and maintaining the ecological balance of Hulun Buir Prairie and even the ecological security of north China, which makes it an important part of the ecological barrier in north China. Lake Hulun is a typical eutrophic steppe lake with poor water quality, serious water eutrophication, and frequent summer algal blooms [15].

### 2.2. Sample Collection

A total of 45, 54, 60, and 60 sampling sites (Figure 1) were set in the study area before the icebound period in autumn (at the beginning of October 2018), the icebound period in winter (at the beginning of March 2019), the melt period in spring (at the beginning of May 2019), and summer (at the end of July 2019), respectively. Due to the low temperature in autumn and winter and the icing in winter,

sampling is difficult and risky, thus the sampling points are fewer than those in summer and spring. The exact location of the sampling point is recorded by GPS (Global positioning system) with an accuracy of 3 m. The overlying water samples in winter were collected after removing the ice with an ice drill. Overlying water samples were collected in 250 mL polyethylene tubs, which were soaked in (1 + 1) HCl for 24 h and then washed clean with ultrapure water and stored in incubators with ice. The samples were sent to our field station ~2 h after all the sampling works were done. The water samples were stored at 4 °C in the refrigerator and analyzed within 24 h.

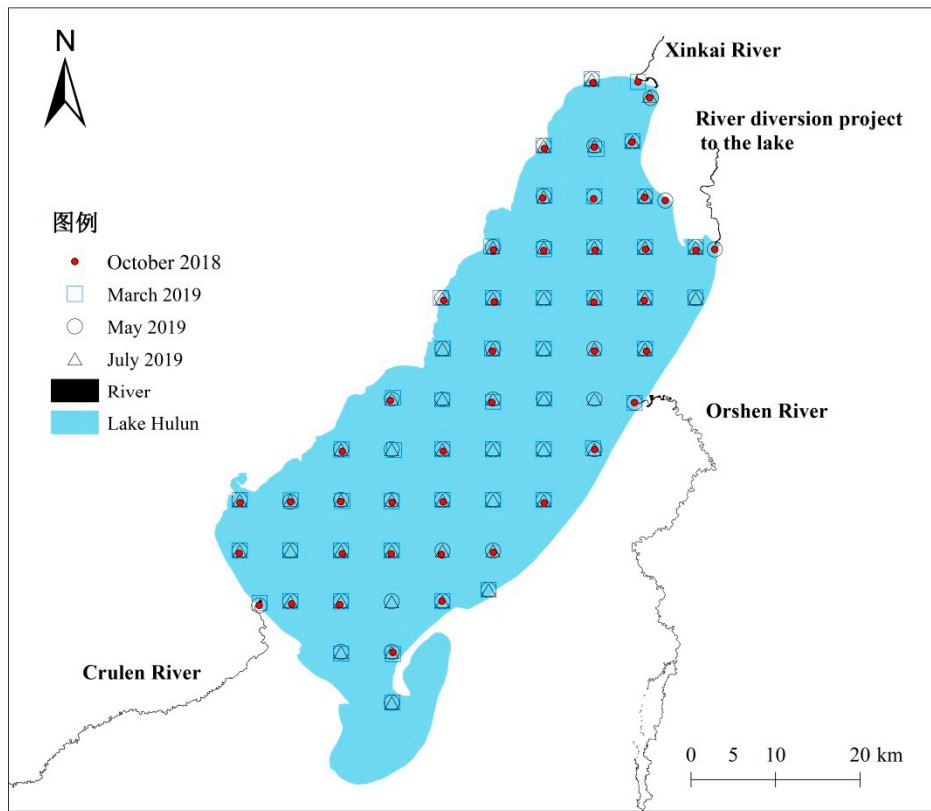

**Figure 1.** Location of the sampling sites.

## 2.3. Sample Analysis

### 2.3.1. Basic Physical and Chemical Index Analysis

Water temperature (T), pH, dissolved oxygen (DO), and electric conductivity (EC) were analyzed by using a multiparameter water quality analyzer (YSI 6600, YSI, Yellow Springs, OH, USA) in the field during sampling. The water transparency (SD) was determined by a secchi disk. Total nitrogen (TN) was analyzed by using the alkaline potassium persulfate digestion–ultraviolet (UV) spectrophotometric method [19]. Ammonia nitrogen ($NH_4^+$-N) was analyzed by using Nessler's reagent spectrophotometry method [20]. Total phosphorus (TP) was analyzed by using the ammonium molybdate spectrophotometry method [21]. Permanganate index ($COD_{Mn}$) was analyzed by using the titrimetric method [22]. Chemical oxygen demand (COD) was measured by a COD detector (DR1010, HACH, Loveland, CO, USA). Salinity was measured by a portable salinity meter (CON450 Meter, Thermo, Waltham, MA, USA). The concentration of chl a was analyzed by performing the spectrophotometry method [23]. The concentration of DOM was represented by DOC and the unit was mg C/L. DOC was analyzed by a TOC analyzer (TOC-V, SHIMADZU, Kyoto, Japan) after the water was filtered using pre-combusted (450 °C for 5 h) glass fiber filters (0.7 μm, GF/F, Whatman).

### 2.3.2. Chromophoric DOM Absorption Analysis

The water samples were filtered using pre-combusted (450 °C for 5 h) glass fiber filters (0.7 μm, GF/F, Whatman). Chromophoric DOM (CDOM) absorption spectra were determined using a Shimadzu UV-2600PC UV-Vis spectrophotometer fitted with a 1 cm quartz cuvette in the spectral region between 190 and 800 nm at 1 nm intervals. CDOM spectra for Milli-Q water were used as a reference. In this study, the absorption coefficient of CDOM at 254 nm ($\alpha_{254}$, m$^{-1}$) was used as a proxy for CDOM concentration. The value of $\alpha_{254}$ was determined using the following equation [24]:

$$\alpha_{254} = 2.303A_{254}/L, \tag{1}$$

where $A_{254}$ is the absorbance at 254 nm, and L is the path length of the cuvette (0.1 m).

### 2.3.3. Fluorescence EEMs-PARAFAC Analysis

Fluorescence absorption of fluorescent DOM (FDOM) spectra was determined using a Hitachi F-7000 fluorescence spectrometer (Hitachi, Tokyo, Japan) with a 150 W Xe lamp as the excitation light source and a 9PMT voltage of 700 V. The excitation wavelength (Ex) scanning range was 200 to 450 nm, and the emission wavelength (Em) scanning range was between 250 and 600 nm. Both the excitation and emission wavelength increments were set to 2 nm. The slit width was 10 nm, and the scanning speed was 12,000 nm·min$^{-1}$. To ensure the comparability of fluorescence spectrum characteristics, the obtained spectra were corrected after the ultra-pure water blank was removed, thereby reducing the influence of instrument conditions and Raman scattering on fluorescence spectra. The EEM fluorescence data should be preprocessed before PARAFAC analysis; the detailed steps can be found in the method used by Wang et al. [25]. PARAFAC modeling was performed using MATLAB (The MathWorks, Natick, MA, USA) with the DOM Fluor toolbox [26]. The number of components was determined based on a core consistency test and split-half validation [26].

The fluorescence index (FI) is the fluorescence intensity ratio of the fluorescence emission spectrum at 450 and 500 nm with an excitation light wavelength of 370 nm [27]. The humification index (HIX) can be calculated from the peak area of Em, which is 435–480 nm, to the fluorescence peak area of Em, which is 300–345 nm at the excitation wavelength of 255 nm [28]. HIX is often used to represent the degree of humification, which means the DOM's degree of conversion of organic matter to humus.

### 2.4. Statistical Analysis

For each water sample, all of the above experiments were performed in triplicate and expressed as average values with a relative error lower than 5%. Statistical analyses of mean value, standard deviation, and t-test were conducted by using SPSS 19.0. The spatial distribution of the sampling sites (Figure 1) was created by ArcGIS 10.2 (ESRI, Redlands, CA, USA). The spatial distributions of DOM, CDOM and the fluorescence intensity of FDOM were created by using the inverse distance weighted method in Suffer 10.0. Figures about the EMM spectra, relative proportions of the fluorescence components, and the relationship between FI and contribution of terrestrial sources of DOM were created by Origin 9.0 (Electronic Arts, Redwood city, CA, USA).

## 3. Results

### 3.1. Physical and Chemical Characteristics of the Lake Water

In general, significant seasonal differences were observed in the physical and chemical properties of the lake water (Table 1). The average temperature of water in March was only 0.05 °C, and the highest temperature in July was 24.2 °C. The pH value was relatively stable and maintained at approximately 9, and the lake water was alkaline. The water transparency was lowest in July, mainly because this month is part of the summer in the Lake Hulun area, when aquatic organisms were relatively flourishing. During the sampling period, algal bloom occurred, further reducing the water transparency. Lake water

salinity was the highest in May, mainly because some of the mineral ions are concentrated by freezing into the ice, thereby decreasing the salinity of the lake. After melting in May, mineral ions from the ice sheet re-entered the lake within a short period, thereby increasing the lake's salt content. During the rainy season in July, the lake's salt content decreased due to the dilution of precipitation. The lowest concentrations of COD, $COD_{Mn}$, TN, and TP were all found in the icebound period, and the higher values were found in July or May. The average comprehensive nutrition state index ranged from 62.6 to 68.7 during the sampling period, indicating that the lake was a middle eutropher. The degree of eutrophication was the highest in July. Thus, Lake Hulun is a eutrophic lake with a large difference in water temperature during different seasons.

**Table 1.** Basic physical and chemical parameters of the overlying water of Lake Hulun.

| Parameters | October 2018 | March 2019 | May 2019 | July 2019 |
|---|---|---|---|---|
| Water temperature (°C) | 6.45 ± 2.04 | 0.05 ± 0.08 | 11.18 ± 1.72 | 24.20 ± 2.60 |
| pH | 8.52 ± 0.45 | 9.17 ± 0.29 | 8.68 ± 0.25 | 8.95 ± 0.10 |
| EC (ms·m$^{-1}$) | 102.35 ± 36.53 | 155.83 ± 36.22 | 165.17 ± 20.53 | 143.28 ± 40.08 |
| SD (cm) | 19.0 ± 3.7 | - | 22.5 ± 9.1 | 11.67 ± 2.58 |
| Salinity (mg·L$^{-1}$) | 646.83 ± 198.34 | 762.33±224.04 | 1088.00 ± 160.19 | 874.00 ± 194.59 |
| $COD_{Mn}$ (mg·L$^{-1}$) | 12.24 ± 2.36 | 8.52 ± 3.53 | 12.80 ± 1.70 | 13.47 ± 1.92 |
| $NH_4^+$-N (mg·L$^{-1}$) | 0.19 ± 0.04 | 0.20 ± 0.08 | 0.24 ± 0.13 | 0.16 ± 0.04 |
| TN (mg·L$^{-1}$) | 1.71 ± 0.31 | 1.77 ± 0.55 | 1.85 ± 0.23 | 2.30 ± 0.50 |
| TP (mg·L$^{-1}$) | 0.19 ± 0.05 | 0.11 ± 0.02 | 0.20 ± 0.03 | 0.20 ± 0.08 |
| COD (mg·L$^{-1}$) | 74.30 ± 12.10 | 71.30 ± 26.65 | 81.22 ± 17.63 | 85.40 ± 16.37 |
| chl a (mg·m$^{-3}$) | 5.5 ± 3.2 | 5.8 ± 1.8 | 10.7 ± 0.8 | 19.7 ± 25.4 |

Note: EC, electric conductivity; SD, water transparency; COD, chemical oxygen demand; TN, total nitrogen; TP, total phosphorus; $NH_4^+$-N, ammonia nitrogen; $COD_{Mn}$, permanganate index.

### 3.2. Spatial and Temporal Distribution Characteristics of DOM

The DOM concentration in the lake water of Lake Hulun had seasonal and spatial differences. The DOM concentration ranged from 6.46 to 42.87 mg C·L$^{-1}$, and the average DOM concentration in different seasons ranged from high to low as follows: summer > spring > autumn > winter. The DOM concentration was highest in summer (32.30 ± 4.62 mg C·L$^{-1}$) and lowest in winter (26.08 ± 11.13 mg C·L$^{-1}$). A significant difference was found in DOM concentration in different seasons ($p < 0.01$); the difference in DOM concentrations in summer and winter was the most significant, and temperature was likely the most important influencing factor. Winter is the icebound period of Lake Hulun, and ice cover has a good isolation effect on exogenous pollution. By contrast, the activity of aquatic organisms is relatively higher at a higher temperature, and DOM will be released during the process of metabolism. The dual contributions of exogenous and endogenous sources resulted in a significantly higher DOM concentration in summer than in winter.

The DOM concentration in lake water also presented spatial differences during different seasons (Figure 2). The spatial difference in DOM concentration at different sampling sites was greatest in winter, with the DOM in the center of the lake being lower than that near the shore. The ice cover formed in winter isolated the water from the air, thus, the water was not affected by exogenous pollution sources. The hydrodynamic force is weak under the ice cover in winter, and probably decreased the water–rock interaction and evaporation. In addition, the sharp decrease in the flow of inflow rivers during the icebound period resulted in significant differences in DOM concentration at different points. The differences in water depth and thickness of ice further led to a larger concentration effect of DOM near the shore than in the lake center. Thus, the DOM concentration in the water near the shore was higher than that in the lake center. The spatial distribution of DOM concentrations in water were similar in summer, spring, and autumn. The DOM concentrations in inlet estuaries, such as the Crulen, Orshen, and Hailar rivers, were relatively low because of the good dilution effects of the inflow rivers on DOM concentration in the lake body.

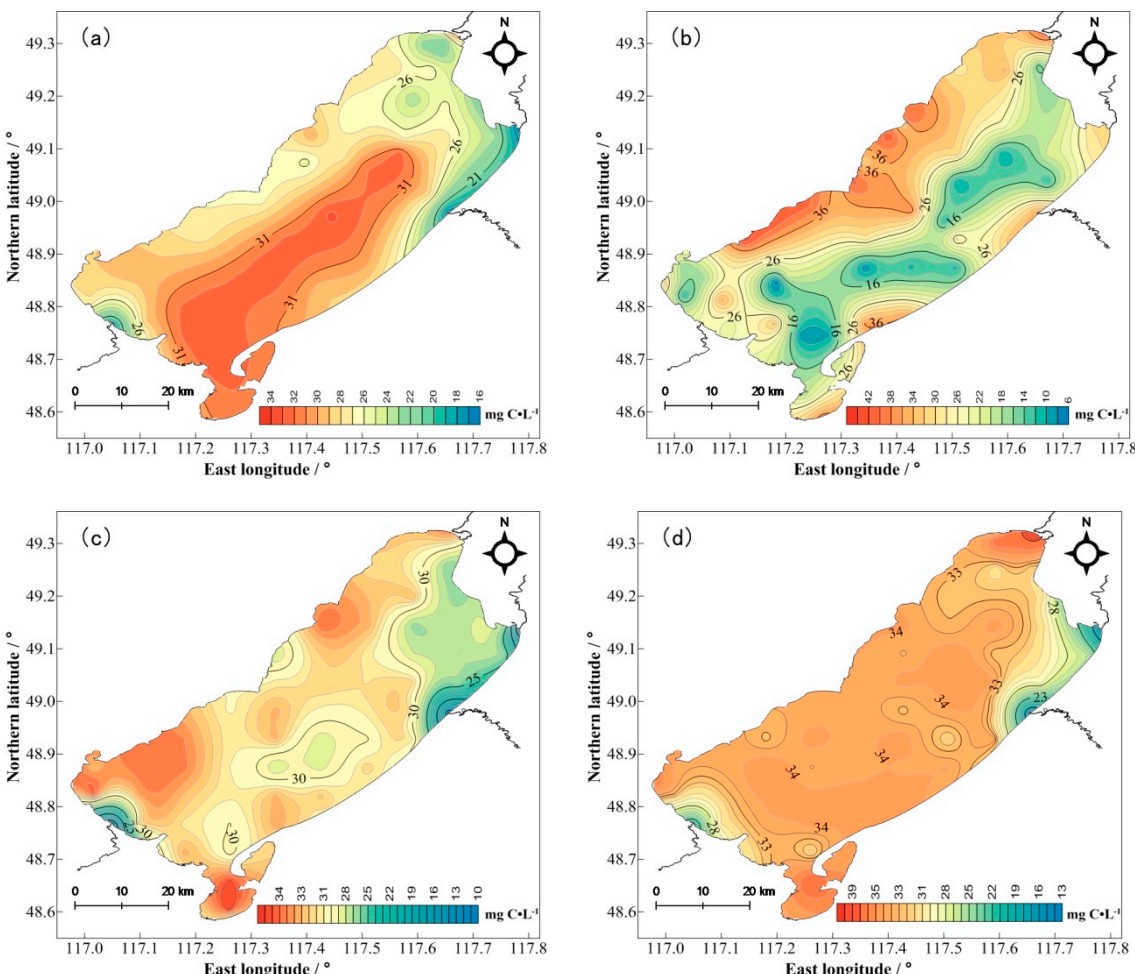

**Figure 2.** Spatial distributions of dissolved organic matter (DOM) concentrations (dissolved organic carbon (DOC), mg C·L$^{-1}$) in (**a**) autumn: October 2018, (**b**) winter: March 2019, (**c**) spring: May 2019 and (**d**) summer: July 2019.

### 3.3. Spatial and Temporal Distribution Characteristics of CDOM

CDOM is part of the DOM and is involved in the absorption of UV and photosynthetically available radiation [29–31]. CDOM is a major component of DOM in water and plays a vital role in carbon cycling in inland water [24,32]. The CDOM concentration, represented by $\alpha_{254}$ in the overlying water of Lake Hulun, ranged from 6.97 m$^{-1}$ to 86.89 m$^{-1}$, and showed significant seasonal differences. The CDOM concentration in winter was 40.69 ± 18.54 m$^{-1}$, which was significantly lower than that in other seasons ($p < 0.01$). The CDOM concentration in summer was the highest, with an average value of 58.80 ± 7.07 m$^{-1}$. The CDOM in Lake Hulun also showed different spatial differences during different seasons (Figure 3). In winter, the CDOM concentration at each sampling point in the water body showed the greatest difference, exhibiting a spatial trend in which the values in the lake center were lower than those near the shore. The spatial distribution characteristics of CDOM in summer and autumn were similar, showing the following high-to-low trend: Crulen River estuary > lake body > Xinkai River estuary > Orshen River and Hailar River estuaries. In spring, the CDOM concentration in the water showed the following distribution trend: lake body > outflow river estuaries > inflow river estuaries.

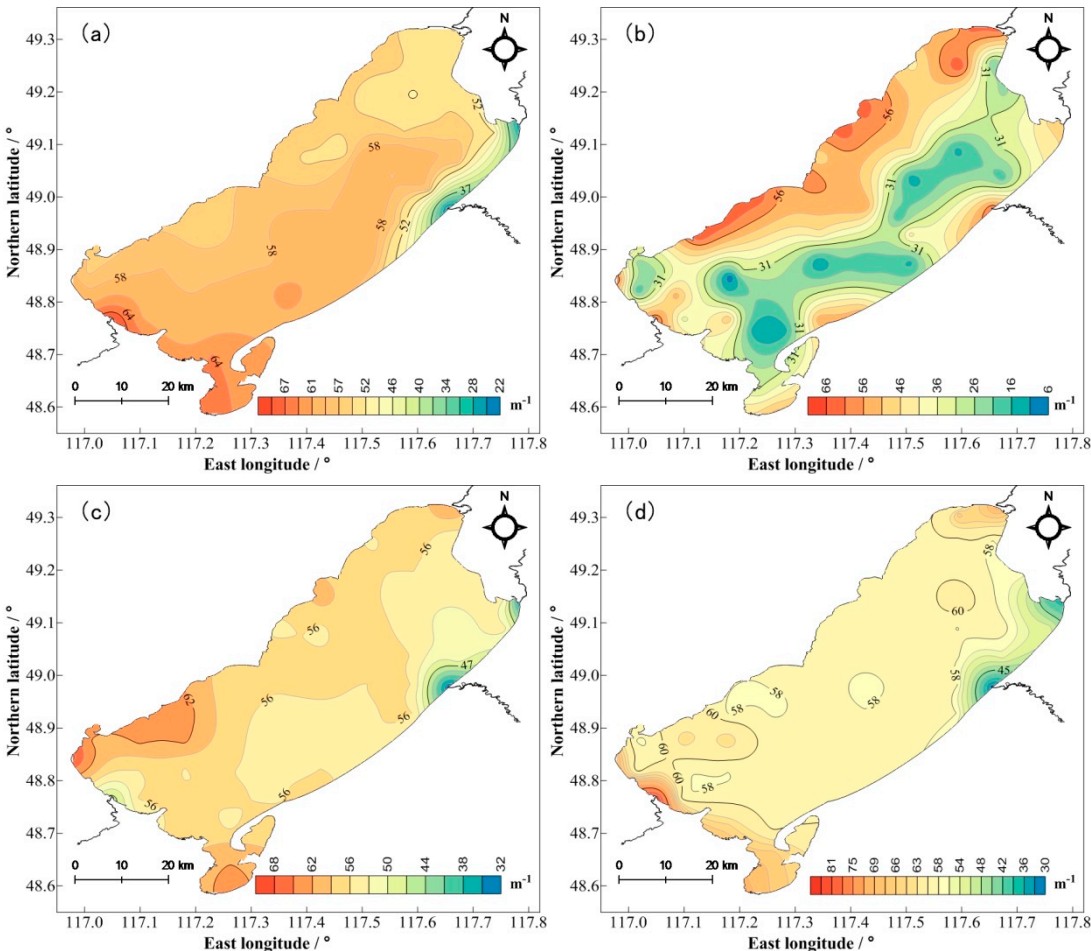

**Figure 3.** Spatial distributions of chromophoric DOM (CDOM) concentrations ($\alpha_{254}$, m$^{-1}$) in (**a**) autumn: October 2018, (**b**) winter: March 2019, (**c**) spring: May 2019, and (**d**) summer: July 2019.

### 3.4. Fluorescence Characteristics and Components of DOM

Apart from CDOM, part of the DOM fraction is referred to as FDOM based on the emission of fluorescence photons after radiation absorption; both CDOM and FDOM are responsible for the optical properties of DOM [30,33]. The EMM fluorescence spectra of FDOM in the overlying water at different sampling sites during different seasons of Lake Hulun were similar. The PARAFAC model was used to analyze the EMM fluorescence data of FDOM, and four fluorescence components with a single maximum emission wavelength were obtained. The EMM fluorescence spectra of the principal components are shown in Figure 4.

A detailed description of the four identified components and examples of matching components in the Open Fluor database are shown in Table 2. Components C1, C2 and C3 matched very well with the previously reported humic-like components derived from terrestrial sources. Component C4 could be a combination of humic-like and tryptophan-like substances, which are associated with biological production in surface water [34,35]. The fluorescence intensity of the components was in the following order: C2 > C3 > C1 > C4. The fluorescence intensity of the components showed two types of seasonal trends: C1 and C2, autumn > summer > spring > winter; and C3 and C4, summer > spring > autumn > winter. The proportion of each component in the total fluorescence intensity also showed a certain seasonal variability. The proportions of C1 and C2 in autumn and winter were higher than those in spring and summer, whereas the proportions of C3 and C4 were the opposite.

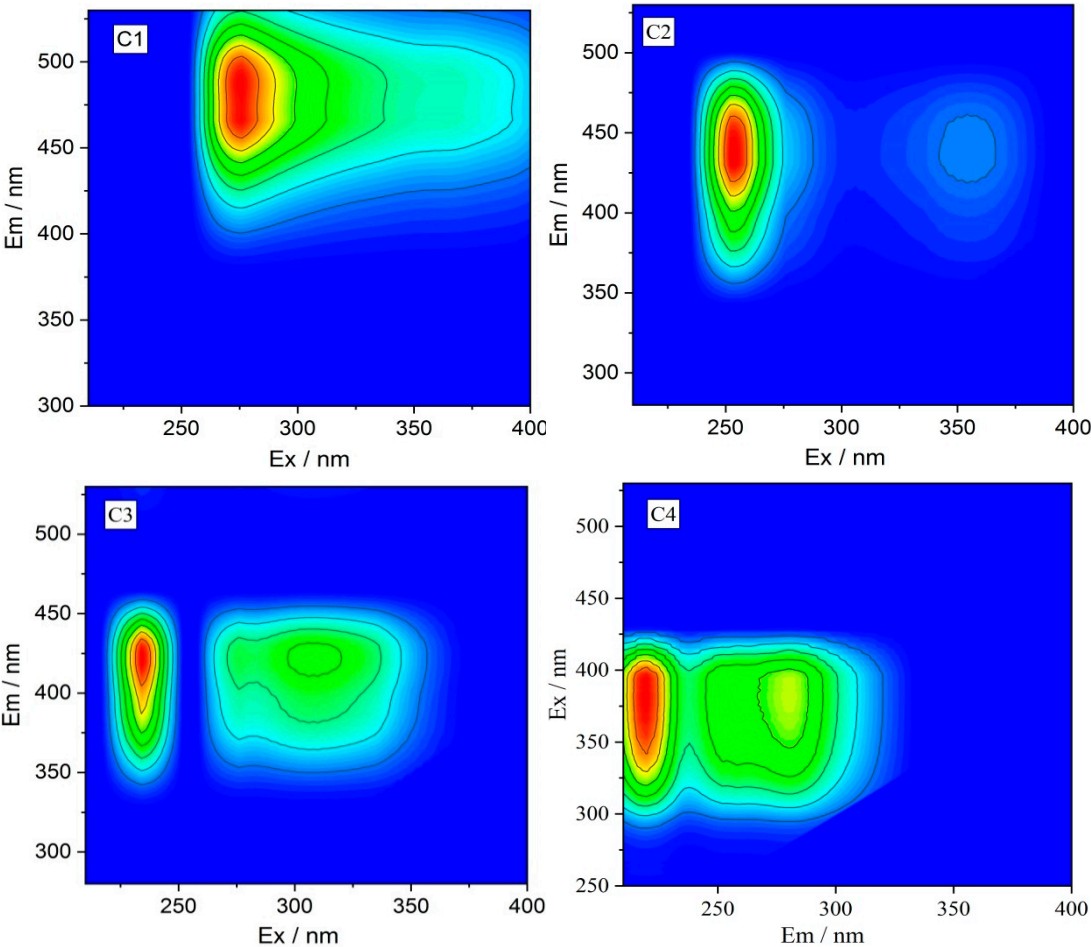

**Figure 4.** EMM spectra of the fluorescence components C1, C2, C3, and C4 of fluorescent DOM (FDOM) in overlying water of Lake Hulun. Ex, excitation wavelength; Em, emission wavelength.

**Table 2.** Excitation and emission maxima of the fluorescent components of FDOM identified in the overlying water of Lake Hulun by parallel factor (PARAFAC) modeling. FDOM, fluorescent dissolved organic matter.

| Component | $Ex_{max}$ (nm) | $Em_{max}$ (nm) | Comparison with other Studies in Open Fluor Database | Description |
|---|---|---|---|---|
| C1 | 260 | 488 | C2: $Ex_{max}$ = 275(340), $Em_{max}$ = 484 [36]; C1: $Ex_{max}$ = 270(370), $Em_{max}$ = 478 [37] | Terrestrially derived humic-like substances with high molecular weight |
| C2 | 254, 356 | 440 | C2: $Ex_{max}$ = 325, $Em_{max}$=430 [38]; C3: $Ex_{max}$ = 275 (345), $Em_{max}$ = 436 [36] | Terrestrially derived humic-like substances |
| C3 | 234, 320 | 420 | C3: $Em_{max}$ = 430 [39]; C2: $Ex_{max}$ = 305, $Em_{max}$ = 390 [40] | Terrestrial humic-like substances |
| C4 | 220, 280 | 384 | C5: $Ex_{max}$ = 280(<240), $Em_{max}$ = 368 [35]; C3: $Ex_{max}$ = 275, $Em_{max}$ = 384 [34] | A possible mix of humic-like and tryptophan-like substances produced as a result of biological production |

Note: Ex, excitation wavelength; Em, emission wavelength.

The total fluorescence intensity of FDOM was 0.27–6.11 R.U., with the significant seasonal variability ($P < 0.01$) showing as summer (2.58 ± 0.49 R.U.) > spring (2.45 ± 0.22 R.U.) > autumn (2.40 ± 0.27 R.U.) > winter (1.64 ± 0.80 R.U.). The spatial distributions of the total fluorescence intensity of FDOM in different seasons were also different (Figure 5). In autumn and spring, the total fluorescence intensity decreased from the southwest to the northeast of the lake area. In winter, the total fluorescence intensity of FDOM in the lake center was lower than that near the shore. In summer, the overall

difference of the total fluorescence intensity of FDOM was not significant, and the highest and lowest values were found at the Crulen and Orshen River estuaries, respectively.

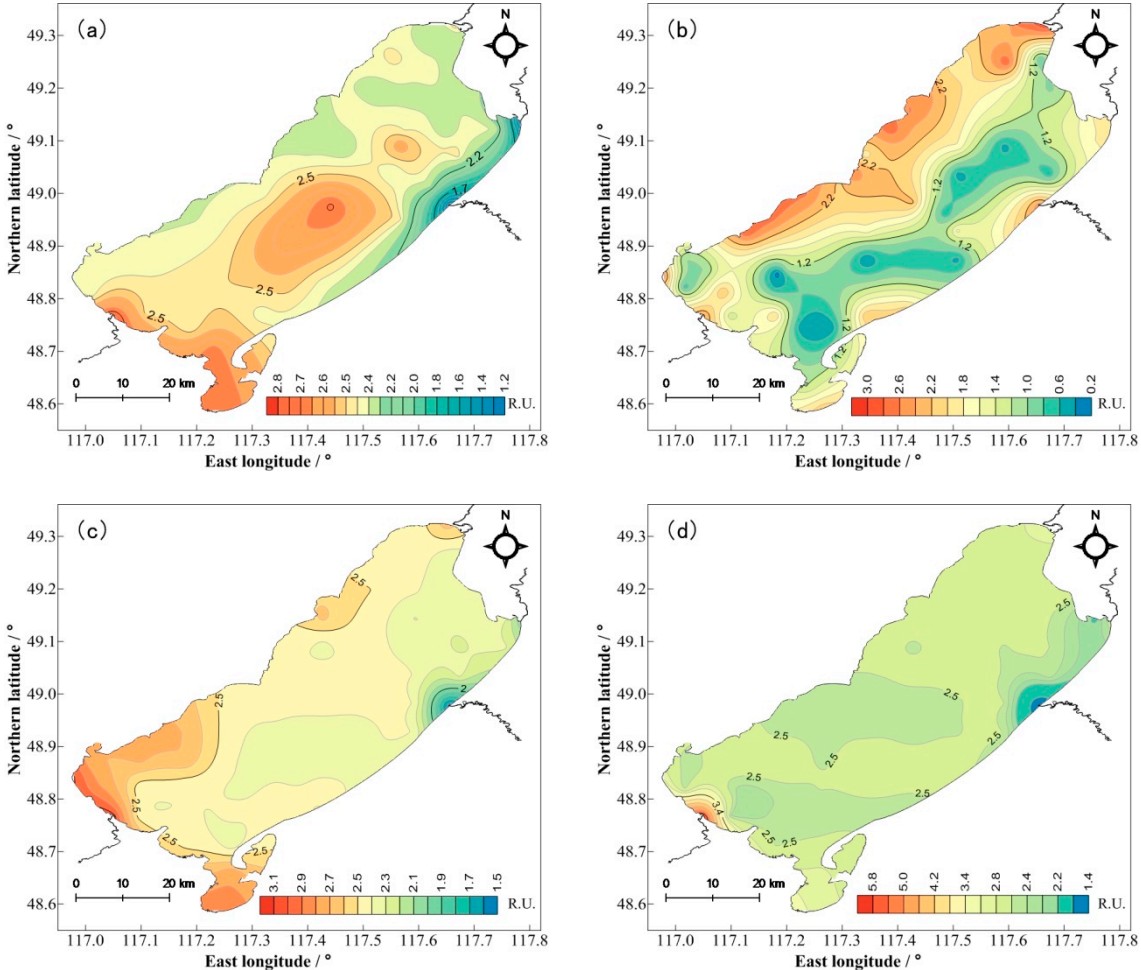

**Figure 5.** Spatial distributions of the total fluorescence intensities of FDOM in (**a**) autumn: October 2018, (**b**) winter: March 2019, (**c**) spring: May 2019, and (**d**) summer: July 2019.

In the overlying water of Lake Hulun, the fluorescence components of DOM were mainly humus-like components (C1, C2, and C3) with a relatively high molecular weight and difficult biodegradation (Figure 6). The humic components accounted for 80.3%–81.4% of the total fluorescence intensity. By contrast, the humic-like and tryptophan-like mixed component (C4) with a low molecular weight and easy biodegradation accounted for the smallest proportion (Figure 6), ranging from 18.6% to 19.7%, with an average of 19.1%. The HIX of the DOM in the water of Lake Hulun ranged from 2.22 to 9.52, with a mean value of 4.22 ± 0.29 for spring, 4.25 ± 0.94 for summer, 4.58 ± 0.51 for autumn, and 4.51 ± 1.20 for winter. The mean values of HIX were between 4 and 10, indicating a small proportion of autochthonous DOM and a high degree of humification [28].

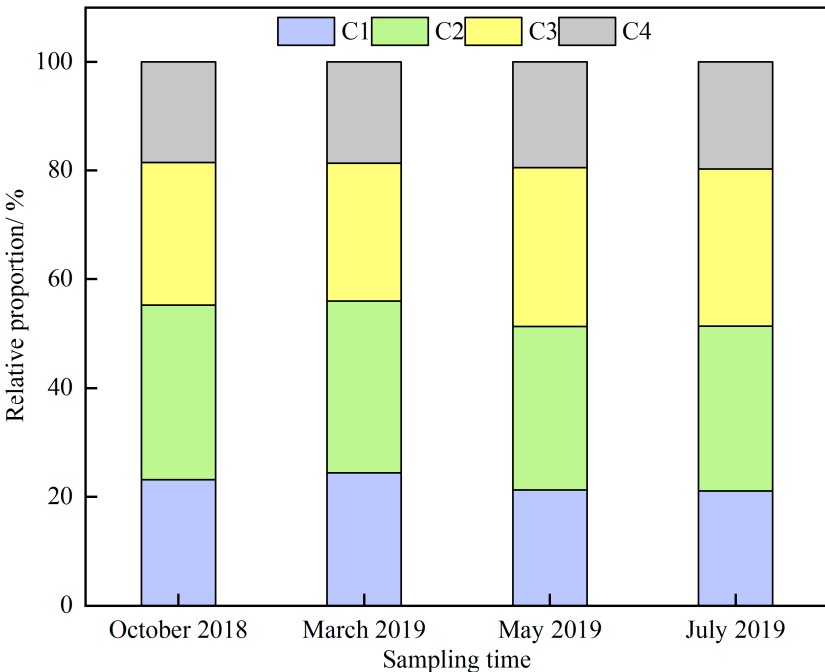

**Figure 6.** Relative proportion of the fluorescence components of DOM in the overlying water of Lake Hulun.

## 4. Discussion

### 4.1. Comparative Analysis of DOM Concentration in Lake Hulun and Other Lakes

To obtain a general understanding of the DOM concentration levels in the water of Lake Hulun in the global context, 18 lakes that are representative of the five lake regions in China and 16 lakes in east Antarctica, Argentina, North America, Japan, and Switzerland were investigated (Table 3). A comparison showed that the DOM concentration in Lake Hulun was significantly higher than that in all of the investigated foreign lakes. The DOM concentration in the water bodies of the investigated foreign lakes ranged from 0.401 mg C·L$^{-1}$ to 4.04 mg C·L$^{-1}$. The lakes in Argentina were all oligotrophic, and the DOM concentrations in the water bodies were the lowest. The average DOM concentration in the waters of the Laurentian Great Lakes (namely, Lake Superior, Lake Huron, Lake Michigan, Lake Erie, Lake Ontario, and Lake Saint Clair) was 2.00 mg C·L$^{-1}$, which is only 1/16 of that of Lake Hulun.

The concentration range of DOM in the investigated lakes in the five lake regions in China was 0.013–11.52 mg C·L$^{-1}$ in the eastern lake region (ELR), 5.11–15 mg C·L$^{-1}$ in the northeastern lake region (NLR), 10.33–34.54 mg C·L$^{-1}$ in the Yungui Plateau lake region (YGR), 3.1–17.2 mg C·L$^{-1}$ in the Tobetam-Qinghai Plateau lake region (TQR), and 13.16–57.11 mg C·L$^{-1}$ in the Inner Mongolia-Xinjiang lake region (MXR). The DOM concentrations in the lakes of MXR are significantly higher than those in other lakes abroad. The DOM concentration in Lake Hulun is higher than that in most lakes in the other domestic lake regions except for MXR, but it is lower than that in Lake Erhai. Compared with other lakes in the MXR, the DOM concentration in Lake Hulun is lower than those in Lake Wuliangsuhai and Lake Daihai and higher than that in Lake Bosten.

The DOM concentration in lake water is affected by a variety of factors, including climatic conditions in the lake basin, the intensity of human activities, land use type, hydrogeology, latitude, and lake characteristics, such as water volume and hydraulic retention time [30,41–43]. The investigated lakes in Argentina are located in the Andes Mountains. Their basins are mostly mountainous and less affected by human activities [44]. The lakes are freshwater and oligotrophic, thus, their DOM concentrations are relatively low. In the same region, DOM concentration may be inversely proportional to the altitude, such as the Laurentian Great Lakes in North America where the DOM concentrations

show an increasing trend at lower altitudes [45]. The main reason for this result is that the DOM of the agricultural and current input is relatively higher at low latitudes. However, despite the large surface area of the Laurentian Great Lakes, which is the largest freshwater lake group in the world, its DOM concentration is not the highest in the world.

The lakes in the TQR in China are located at high altitudes and are mainly supplied by glaciers and rainfall [42,46]. Moreover, the intensity of human influence activities is relatively low. In addition, the DOM in their water bodies is prone to photodegradation given the strong light intensity and UV radiation [42], resulting in the lowest DOM concentration in the entire country. The ELR and YGR are also two distinctive lake regions in China that have highly intense human activity and serious eutrophication [24,41]. The DOM concentration in the ELR was lower than that in the YGR, and it is mainly related to factors such as the lake's characteristics, hydrodynamic conditions, and pollution sources. On the one hand, the water surface areas of lakes in the ELR are significantly larger than that of lakes in the YGR, thereby they have a good dilution effect on DOM. On the other hand, water retention time is one of the important reasons for the difference in DOM between the two lake regions. The water retention time of the investigated lake in the ELR ranged from 27 to 127 days, whereas the value was 485–891 days for the YGR [47]. A long water retention time is conducive to the accumulation of DOM in water bodies. The DOM concentrations in lakes of the MXR are higher than the other compared lakes in this paper, which is mainly due to climatic conditions, altitude, and lake conditions. The MXR is located in the high-latitude region of China, and lakes are mostly the last resting place of inland water systems [24]. At the same time, the lakes are concentrated, the lake areas are decreased, and the water is shallow because the evaporation is much greater than the excess supply. Most of the lakes in the MXR develop into saltwater lakes or salt lakes, which also leads to high DOM concentrations.

**Table 3.** Comparison of DOM concentration in Lake Hulun and other lakes.

| Country/Region | Lake | Lake Type | Nutrition Status | Altitude (m) | Area (km$^2$) | Water Depth (m) | DOC (mg C/L) | Sampling Time | Reference |
|---|---|---|---|---|---|---|---|---|---|
| China | Lake Poyang | Shallow freshwater lake | Light eutropher | 1–12 | 3886 | 12–20 | 3.71 ± 1.65 | 2013.6 | [48] |
| | Lake Taihu | Shallow freshwater lake | Light eutropher | 1.1 | 2338 | 1.9 | 2.19 ± 0.99 | 2010.2 | [49] |
| | Lake Hongze | Shallow freshwater lake | Light eutropher | >13 | 1960 | 1.77 | 13.9 ± 0.7 | 2016.4 | [29] |
| | Lake Chao | Shallow freshwater lake | Light eutropher | 5–10 | 770 | 3.0 | 0.013 ± 0.001 | 2013.3–2015.4 | [50] |
| | Lake Baiyangdian | Shallow freshwater lake | Light eutropher | 4.1–4.6 | 336 | - | 11.52 ± 2.95 | 2012.6 | [51] |
| NLR | Lake Chagan | Shallow freshwater lake | Light eutropher | 126 | 375.2 | 2.3 | 15 | 2011, 2014, 2015 | [30] |
| YGR | Lake Tiancai | Alpine freshwater lake | Oligotropher | 3880 | 0.2 | 6.5* | 10.33 | 2013.6 | [5] |
| | Lake Dianchi | Plateau freshwater lake | Light eutropher | 1886 | 298 | 4.1 | 15.15 | 2013 | [52] |
| | Lake Erhai | Plateau freshwater lake | Mesotropher | 1974 | 256.5 | 15 | 34.54 ± 4.26 | 2016.4 | [41] |
| TQR | Lake Qinghai | Plateaus, saltwater lake | Light eutropher | 3199 | 4349.9 | 21 | 17.2 | 2014.9 | [42] |
| | Lake Yangzhuoyongcuo | Plateaus saltwater lake | Oligotropher | 4447 | 571.4 | 20–40 | 8.1 | 2015.6 | [42] |
| | Lake Keluke | Plateau freshwater lake | Mesotropher | 2817 | 54.7 | 5 | 5.8 | 2014.9 | [42] |
| | Lake Bangongcuo | Plateau freshwater lake | Oligotropher | 4244 | 451.1 | 57 | 3.1 | 2015.7 | [42] |
| MXR | Lake Bosten | Inland freshwater lake | Mesotropher | 1048 | 1000 | 9 | 13.16 ± 3.63 | 2015.6 | [8] |
| | Lake Wuliangsuhai | Shallow freshwater lake | Eutropher | 1018 | 300 | 0.5–1.5 | 57.11 ± 7.35 | 2009.11 | [53] |
| | Lake Daihai | Inland brackish lake | Eutropher | 1220 | 55 | 4 | 44.56 ± 2.44 | 2018 | Unpublished research results of the research team |
| | Lake Hulun | Inland brackish lake | Eutropher | 545.6 | 2339 | 5.75 | 32.30 ± 4.62 | 2019.7 | This paper |

**Table 3.** *Cont.*

| Country/Region | Lake | Lake Type | Nutrition Status | Altitude (m) | Area (km$^2$) | Water Depth (m) | DOC (mg C/L) | Sampling Time | Reference |
|---|---|---|---|---|---|---|---|---|---|
| Antarctica | Lake Richardson | Freshwater lake | Oligotropher | - | 5.44 | - | 1.16 | 2017.2 | [54] |
| Argentina | Lake Schmoll | Shallow lake | Oligotropher | 1950 | 0.028 | 5 * | 0.401 ± 0.000 | 2013–2015 | [44] |
| | Lake Toncek | Shallow lake | Oligotropher | 1750 | 0.05 | 9 * | 0.732 ± 0.027 | 2013–2015 | [44] |
| | Lake Juventus | Shallow lake | Oligotropher | 1010 | 0.046 | 12.0 * | 2.079 ± 0.007 | 2013–2015 | [44] |
| | Lake Morenito | Shallow lake | Oligotropher | 770.8 | 0.365 | 10.5 * | 1.45 | 2019 | [44] |
| | Lake Moreno | Deep lake | Oligotropher | 770.8 | 6.1 | 90–106 * | 0.73–0.74 | 2019 | [44] |
| | Lake Escondido | Shallow lake | Oligotropher | 772.8 | 0.09 | 8.3 * | 4.04–4.29 | 2019 | [44] |
| America, Canada | Lake Superior | Freshwater lake | Oligotropher | 180 | 82000 | 147 | 1.188 | 2013.9 | [45] |
| Canada | Lake Huron | Freshwater lake | Oligotropher | 177 | 59600 | 60 | 1.872 | 2013.8 | [45] |
| America | Lake Michigan | Freshwater lake | Oligotropher | 177 | 58016 | 85 | 2.1936 | 2013.8 | [45] |
| America, Canada | Lake Erie | Freshwater lake | Eutropher | 174 | 25744 | 19 | 2.574 | 2013.9 | [45] |
| | Lake Ontario | Freshwater lake | Oligotropher | 85 | 19554 | 86 | 2.337 | 2013.9 | [45] |
| America, Canada | Lake Saint Clair | Freshwater lake | Eutropher | 175 | 1210 | 7.8 | 1.848 | 2013.8 | [45] |
| Japan | Biwa Lake | Freshwater lake | Mesotropher | 84.371 | 670.25 | 41.2 | 1.47 | 2006 | [55] |
| Switzerland | Lake Bienne | Freshwater lake | - | 429 | 39.50 | 74 * | 3.16 | 1987–2010 | [56] |
| | Lake Constance-Obersee | Freshwater lake | - | 395 | 473.00 | 254 * | 1.3 | 2000–2009 | [56] |

Note: * the maximum water depth; ELR, the eastern lake region in China; NLR, the northeastern lake region in China;YGR, the Yungui Plateau lake region in China; TQR, the Tobetam-Qinghai Plateau lake region in China; MXR, the Inner Mongolia-Xinjiang lake region in China.

### 4.2. Source Apportionment of DOM in Overlying Water of Lake Hulun

The sources of organic matter in lake water are generally divided into terrestrial and autochthonous sources [57,58]. The terrestrial DOM mostly comes from the input of organic matter caused by soil, forest, animal and plant residues, and human activities in the basin. Terrestrial DOM usually has high C:N ratios over 15 [14,59] and humus-like materials are the main component [57]. Autochthonous DOM is mainly produced by the degradation and secretion of submerged plants, algae, bacteria, and microorganisms that are mainly composed of protein-like substances [60]. EMM fluorescence spectroscopy is a successful method to identify the source of organic matter in all kinds of water bodies [14]. FI is a good metric for differentiating DOM from terrestrial and autochthonous sources. Generally, an FI of DOM of approximately 1.40 indicates that the fluorescence emitting groups of DOM are mainly from terrestrial sources; an FI of approximately 1.90 indicates that the fluorescence emitting groups of DOM are mainly from autochthonous sources [27]. C:N has also been widely used in the identification of DOM sources in water. C:N and FI were used to identify the source of DOM in Lake Hulun.

The C:N ratios of the DOM in the overlying water of Lake Hulun ranged from 6.15 to 51.37, with a mean value of 19.98, 18.64, 20.20 and 21.01 for spring, summer, autummn and winter, respectively. The mean value of C:N was bigger than 15, which means that the DOM is mainly derived from a terrestrial source. The FI of the DOM in the water of Hulun Lake ranged from 1.33 to 1.67, with a mean value of $1.46 \pm 0.02$ for spring, $1.47 \pm 0.02$ for summer, $1.44 \pm 0.03$ for autumn, and $1.40 \pm 0.03$ for winter. The FI values all fluctuated at approximately 1.4, indicating that the DOM mainly came from terrestrial sources formed by the decomposition of higher plants and animal residues or the leaching of soil humus, among others; this finding is similar to that of Chen et al. [14]. The result for the DOM source identified by FI is also similar to that found by the C:N ratio.

To further understand the relative contributions of terrestrial and autochthonous sources of DOM in the lake, the corresponding relationship between FI and the relative proportion of terrestrial sources was established to estimate the contribution of terrestrial and autochthonous sources (Figure 7). The relationship was established by referring to the relationship curve between FI and aromaticity of DOM ($y = 3.94x^{-0.316}$) [27] on the theoretical basis that the change between the two types of metadata for FI and aromaticity are correlated and not affected by water pH. Results showed that the relative contributions of terrestrial sources to DOM in spring, summer, autumn, and winter were $72.79\% \pm 5.61\%$, $70.61\% \pm 6.95\%$, $80.11\% \pm 6.91\%$, and $91.41\% \pm 7.91\%$, respectively, indicating that the terrestrial sources make an absolutely dominant contribution to the DOM in the overlying water of Lake Hulun.

The DOM concentration in the overlying water of Lake Hulun was relatively higher than most of the compared lakes, and it was significantly higher than that of Lake Taihu, a typical eutrophic lake in China. Previous research has shown that the self-derived tryptophine-like component was the main component of DOM in the water of Lake Taihu [49], and the DOM was highly bioreactive [60]. Given that the DOM concentration in Lake Hulun was several times that of Lake Taihu, the question now is whether the bioavailability of the DOM is as high as or even higher than that of Lake Taihu. Our study results showed that the DOM in the overlying water of Lake Hulun was mainly derived from terrestrial sources and dominated by humic-like components with high molecular weight and a high degree of humification. Hence, the bioavailability of the DOM in Lake Hulun might be not as high as expected. However, this paper is just a preliminary analysis, and more scientific and reasonable experiments, such as biodegradation experiments are needed to better understand the bioavailability of DOM in the future.

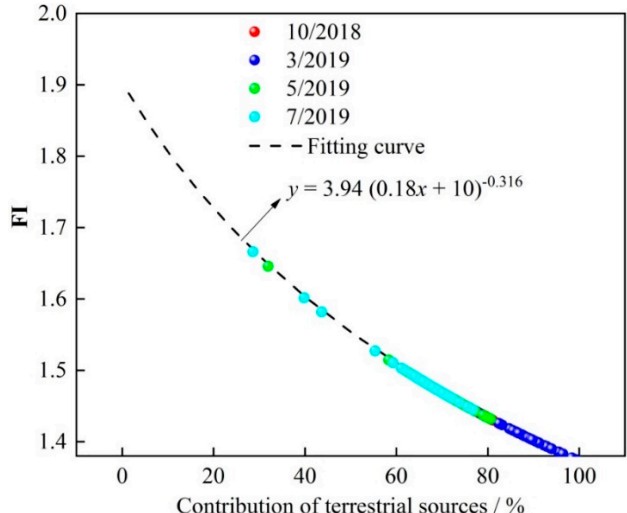

**Figure 7.** Relationship between the fluorescence index (FI) and contribution of terrestrial sources of the DOM in the overlying water of Lake Hulun.

## 5. Conclusions

The DOM in Lake Hulun has a high concentration level ranging from 6.46 to 42.87 mg C/L. Both temporal and spatial differences in DOM concentration were found. The concentration was highest in summer and lowest in winter. The spatial distributions of DOM in spring, summer and autumn were similar and showed a trend whereby DOM was lower at inlet estuaries and the DOM concentration in the water near the shore was higher than that in the lake center in winter. The concentration of CDOM ranged from 6.97 to 86.89 $m^{-1}$ and showed similar temporal and spatial distribution trends as DOM. Four fluorescence components of DOM were detected: three humic-like components C1–C3, and one component C4 had a humic-like and tryptophan-like mixture. The humic-like components took 81.9% of the total fluorescence intensity, which ranged from 0.27 to 6.11 R.U. The mean HIX value was $4.22 \pm 0.29$, $4.25 \pm 0.94$, $4.58 \pm 0.51$ and $4.51 \pm 1.20$ in spring, summer, autumn and winter, respectively, which represent a high degree of humification. The DOM in overlying water of Lake Hulun mainly comes from terrestrial sources.

**Author Contributions:** W.W. wrote the paper and analyzed the data; S.W., X.J. and B.Z. conceived and designed the experiments; W.W., and J.C. performed the experiments. All authors have read and agreed to the published version of the manuscript.

**Funding:** This research was funded by the National Natural Science Foundation of China, grant number 42202018033.

**Acknowledgments:** The authors thank the participants in this study for their time and useful insights. We also thank the support provided by the National Nature Science Foundation of China (No. 42202018033).

**Conflicts of Interest:** The authors declare no conflict of interest.

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
