# Peer review of "Characteristics and Source of Dissolved Organic Matter in Lake Hulun, A Large Shallow Eutrophic Steppe Lake in Northern China"

_water, doi:10.3390/w12040953_

Round 1

Reviewer 1 Report

Manuscript WATER-745716 reports the spatiotemporal variability of dissolved organic matter (DOM) concentration and optical character for a large lake in China. The manuscript requires substantial revision for several reasons: 1) The source and bioavailability of DOM is never explicitly assessed; 2) The methods are inadequately described; 3) The study is not properly introduced or contextualized; 4) The results are not presented appropriately, based on the inadequately described methods; 5) DOM optical characteristics are significantly over-interpreted by the authors; and 6) Major conclusions are stated regarding variables not carefully measured/analyzed, or measured at all, like global climate change, regional environmental change, and land use patterns.

Introduction could use reorganization. Specifically:

  1. A) I don’t understand why the authors highlight, in the very first paragraph, a regionally-specific finding from an unrelated ecosystem - Gors et al. studied an estuary, not a steppe lake. Yet, this study pivots the entire first paragraph toward eutrophication as a topic. This comes off as quite awkward writing.
  2. B) After pivoting from a Baltic estuary to eutrophication in general, the authors then discuss a specific instance in the Mediterranean.
  3. C) Terms are introduced, but never defined (like humification, eutrophication).
  4. D) Some key components of the study are never introduced, like ‘what is EEM-PARAFAC technology?’ and ‘how does this technology give useful information about DOM?’ The acronym, EEM-PARAFAC, is never even defined.
  5. E) A lot of information belongs in the study site section (Third/fourth paragraphs about Lake Hulun).
  6. F) The experimental design is never introduced (what specifically do they mean to ‘understand systematically’?)
  7. G) The connection between their introduced study and global warming is absent at the moment. How are they explicitly assessing global warming’s role in DOM concentration, patterns and character?
  8. I) I would like to see hypotheses – what were the authors expecting?

Methods require greater detail and clarity in expressing exactly what is being measured. Specifically:

  1. A) How were water samples collected during the icebound period? Wasn’t the water beneath the ice?
  2. B) Why were some sampling locations not sampled at certain times (as indicated in Figure 1)? More minor concern: Why isn’t Figure 1 cited anywhere in the text?
  3. C) How long did it take for the water samples to be “sent to the field station”? Were water samples on ice and/or in the dark during the trip?
  4. D) What does “part of the water” mean, quantitatively? And why was only part of the water sample filtered?
  5. E) What pore size was used for filtration? Also, what does “clean plastic tubs” mean, specifically? Please specify the tub materials and dimensions and, more importantly, how they were cleaned and how you know the tubs were clean enough for DOM sample collection.
  6. F) Spectral slope ratios (Sr) and SUVA254 do not represent CDOM molecular weight. Sr can provide a proxy for relative molecular weight comparisons when confirmed by additional analyses for a specific system; however, by itself it may not be a relative molecular weight proxy. SUVA254 is a proxy for aromaticity, not for molecular weight.
  7. G) Bioavailability methods are never described, yet the authors claim in the title, abstract, introduction and discussion that they have information on this. This is a very specific test for DOM quality (usually consisting of incubation experiments). I recommend the authors remove bioavailability from their manuscript as they do not have any measurements of this.
  8. H) Why do the authors say that they have sediment samples in their Statistical Analysis section?
  9. I) How were exact sampling locations returned to? Were there markers? GPS? To what level of geospatial accuracy are the authors certain that they returned to the same sampling location each time?

Results section is generally written ok; however there are a few issues with the presentation. Specifically:

  1. A) The authors present isopleth maps of their analytical parameters (DOC, SUVA, etc), yet drawing such contours is very likely to produce misleading geographic trends due to the evenly-spaced sampling plan and environmental spatial autocorrelation. Please re-design the maps (Figures 2, 3, and 5) to show the observation points rather than a contoured map to more clearly represent their findings.
  2. B) The descriptions of the FDOM components are pretty standard, but the authors could go deeper with this. I recommend the authors compare their results with the Openflour database (https://openfluor.lablicate.com/) to see if these component descriptions consistently match what others have found. It may be that one fluorophore component represents different compounds between different sites.

The discussions and conclusions are far overextended, in my opinion.

  1. A) DOM sources are entirely inferred from fluorescence characteristics, which (i) represent a small fractio of the DOM pool and (ii) could be generated from a large diversity of possible sources that can only be constrained when fluorescence analyses are complemented by other analytical methods (like FTICR-MS or stoichiometric ratios). Thus, the discussion of DOM sources is all highly inferential and uncertain.
  2. B) Section 4.4. (“influence factors”) is entirely “arm waving.” There are statements about many factors not measured in the study: evaporation, pollutants and ‘purification capacity’, natural grassland area, windy weather, herbage cuttings, soil factors… the list goes on. Please delete this section entirely.
  3. C) Bioavailability is never measured in this study.
  4. D) Global climate change, regional environmental change, and land use patterns may very likely be important factors affecting this lake’s DOM amount/optical character. However, this study provides no convincing observational or modelling evidence for any of these variables. Please remove as it is unscientific to make these concluding claims without any robust analysis of the possible mechanisms connecting the DOM observations to these broader processes.

Author Response

We greatly appreciate you  for the positive and constructive comments and suggestions on our manuscript entitled “Characteristics, source and bioavailability of dissolved organic matter in Lake Hulun, a large shallow eutrophic steppe lake in northern China, under global warming.” (ID: water-745716). We have studied the comments carefully and have made revisions that are marked in red in the paper. We have tried our best to revise our manuscript according to the comments. The main corrections in the paper and the responses to the comments are uploaded as an attachment.

Reviewer 2 Report

The authors did a tremendous amount of work and paper was written fairly well. However, there are a few improvements that can be made.

1. Since you collected a good amount of data, you can employ some multivariate statistical techniques and/ or machine learning to improve the quality of the paper. Most of the discussion points are made from either the basic statistics that were carried out or statements that cannot be justified statistically. In addition, you may have collected other fluorescence and absorbance based measurements which you can add in the analyses and improve the manuscript. 

Comments of the abstract. 

The abstract should be reviewed and re-written. Some of the suggestions for the abstract include;

(i) Line 3; The authors wrote that "Lake Hulum has a relatively high concentration " Using the word relative shows that you are comparing it to something or other lakes which is not stated in the sentence.

(ii) line 18: The authors use the word "components", which components did you mean?? PARAFAC? 

(iii) line 19: The authors mention about influencing factors of DOM were discussed. Which influencing factors did the authors mean? Factors influencing the quality or the quantity of DOM?

(iv) lines 19-20: The authors stated the concentration of DOM, in my guess this was DOC, not DOM, DOM has more elements like N, P, S, and many others unknowns that you did not measure.

MAIN TEXT

Line77: Correct 3D EMs -PARAFA to EMM-PARAFAC

Line 180: "The DOM concentration represented by DOC". This statement should appear early in the manuscript to avoid confusion of DOC and DOM.

Line316: Mentioning that the DOM concentrations in lakes of MXR are highest in the world in an overstatement. The world is bigger than the lakes mentioned in the study. 

Line 327; See comment above.

Lines 346-348; The authors talk about the biodegradability of DOM but there were no experiments carried out to assess the biolability of DOM. I don't think biodegradability can only be assessed by PARAFAC components and HI and FI. Biodegradability is assessed by setting up biodegradation experiments. 

Author Response

(The authors gave the same response as above.)

Reviewer 3 Report

Eutrophication of water bodies, a global process of increasing the trophic level of water objects causing an increase in their productivity and deterioration of water quality, is one of the most relevant issues in the modern ecology and hydrobiology.

Hulun Lake located in the steppe areas of China near the borders of Mongolia and Russia requires special attention because here we observe an increasing shortage of water resources and their seasonal lack in a significant part of the region as well as anthropogenic transformation and deterioration of water quality in the transboundary water objects.  

The paper is undoubtedly of high quality and based on a high methodological level. This is an extensive scrupulous study that provides a great array of data perfectly processesed and interpreted, with an exhaustive analysis of the authors’ results and literary information.

I have no comments on the paper and recommend it to be published.

Author Response

Thans for your approval!

Round 2

Reviewer 1 Report

The revised manuscript’s methodological detail, discussion, and writing has been much improved. I would like to see some minor revisions before publication:

1) Please include the methods for spatial interpolation in the statistical analysis section of the methods.

2) I suggest the reference to bioavailability be removed from line 17 of the abstract and from the keywords, as the authors admit that this is not explicitly assessed in their study. The authors treatment of bioavailability throughout the rest of the manuscript is now appropriate.

Author Response

Dear Professor:

    We greatly appreciate you  for the positive and constructive comments and suggestions on our manuscript entitled “Characteristics, source and bioavailability of dissolved organic matter in Lake Hulun, a large shallow eutrophic steppe lake in northern China, under global warming.” (ID: water-745716). We have studied your comments carefully and have made revisions that are marked in blue in the paper. We have tried our best to revise our manuscript according to the comments. The main corrections in the paper and the responses to the comments are as follows:

Response to Reviewer 1 Comments

Point 1: Please include the methods for spatial interpolation in the statistical analysis section of the methods. 

 Response 1: The method for spatial interpolation was supplemented in section 2.4, and the section was reorganized. (line 145-147)

Point 2: I suggest the reference to bioavailability be removed from line 17 of the abstract and from the keywords, as the authors admit that this is not explicitly assessed in their study. The authors treatment of bioavailability throughout the rest of the manuscript is now appropriate.

Response 2: The term “bioavailability”has been removed from line 17 of the abstract and from the keywords according to the comment. The revised portion was marked in blue (line 16, 32)

Reviewer 2 Report

The authors addressed my comments.

Author Response

Dear Professor;

   We greatly appreciate you for the positive and constructive comments and suggestions on our manuscript entitled “Characteristics, source and bioavailability of dissolved organic matter in Lake Hulun, a large shallow eutrophic steppe lake in northern China, under global warming.” (ID: water-745716).

  Best wishes!